# Weak Ultraviolet B Enhances the Mislocalization of Claudin-1 Mediated by Nitric Oxide and Peroxynitrite Production in Human Keratinocyte-Derived HaCaT Cells

**DOI:** 10.3390/ijms21197138

**Published:** 2020-09-27

**Authors:** Mao Kobayashi, Shokoku Shu, Kana Marunaka, Toshiyuki Matsunaga, Akira Ikari

**Affiliations:** 1Laboratory of Biochemistry, Department of Biopharmaceutical Sciences, Gifu Pharmaceutical University, Gifu 501-1196, Japan; 155032@gifu-pu.ac.jp (M.K.); 165041@gifu-pu.ac.jp (S.S.); 136033@gifu-pu.ac.jp (K.M.); 2Education Center of Green Pharmaceutical Sciences, Gifu Pharmaceutical University, Gifu 502-8585, Japan; matsunagat@gifu-pu.ac.jp

**Keywords:** claudin-1, nitric oxide, nitration, mislocalization

## Abstract

A tight junction (TJ) makes a physical barrier in the epidermal cells of skin. Ultraviolet (UV) light may disrupt the TJ barrier, but the mechanism has not been well clarified. Weak UVB (5 mJ/cm^2^) caused mislocalization of claudin-1 (CLDN1), a component of the TJ strand, and disruption of TJ barrier in human keratinocyte-derived HaCaT cells. The UVB-induced mislocalization of CLDN1 was inhibited by monodansylcadaverine (MDC), a clathrin-dependent endocytosis inhibitor, suggesting that UVB enhances the internalization of CLDN1. Transepidermal electrical resistance and paracellular flux of lucifer yellow, a fluorescent hydrophilic marker, were rescued by MDC. UVB changed neither the total nor phosphorylation levels of CLDN1, but it increased both mono-ubiquitination and tyrosine nitration levels of CLDN1. Fluorescence measurements revealed that UVB increased intracellular free Ca^2+^, nitric oxide (NO), and peroxynitrite contents, which were inhibited by Opsin2 (OPN2) siRNA, suggesting that OPN2 functions as a UVB sensor. The effects of UVB were inhibited by an antagonist of transient receptor potential type vanilloid 1 (TRPV1) and Ca^2+^ chelator. Both NO donor and peroxynitrite donor induced the mislocalization of CLDN1 and disruption of TJ barrier, which were rescued by a NO synthase (NOS) inhibitor and a peroxynitrite scavenger. Weak UVB irradiation induced the disruption of TJ barrier mediated by mislocalization of CLDN1 in HaCaT cells. The OPN2/TRPV1/NOS signaling pathway may be a novel target for preventing destruction of the TJ barrier by UVB irradiation.

## 1. Introduction

Skin has a function of maintaining a physical barrier between the inside and outside of the body. The skin is divided into two main structural layers: the epidermis and dermis. The epidermis is the most outer layer that contains keratinocytes and melanocytes. The mammalian epidermis is comprised of four layers: the basal, spinous, granular, and corneum layers [1]. The divided cells in the basal layer have proliferative capacity and move to the next layer. The differentiated keratinocytes, which lack proliferative activity, exist in the spinous and granular layers. The epidermis can be attacked by physical, chemical, and microbial agents on a daily basis [2]. The stratum corneum forms a continuous sheet of corneocytes and serves as the first line of defense. A tight junction (TJ) also plays an important role in the physical barrier in the stratum granulosum and upper stratum spinosum [3].

Human skin keratinocytes form the TJ at the apical side of lateral membrane to connect neighboring cells. The TJ controls the selective paracellular flux of small molecules, electrolytes, and water, whereas it prevents intrusion of pathogens [4]. The TJ is composed of multiprotein complex including transmembrane proteins such as claudins (CLDNs), occludin, junctional adhesion molecules, and tricellulin, and adaptor proteins such as zonula occludens-1 (ZO-1) and ZO-2. CLDNs are four transmembrane proteins that consist of a large family of over 20 subtypes in mammals [5,6]. CLDNs can bind to the postsynaptic density-95/discs large/zonula occludens-1 (PDZ) domain of the scaffolding protein ZO-1 via the carboxy terminal PDZ-binding motif [7]. CLDN1 and CLDN4 are abundantly expressed in human keratinocytes [8]. CLDN1-deficient mice develop aberrant barrier function in the stratum granulosum and die within 1 day [9]. The patients with neonatal ichthyosis and sclerosing cholangitis syndrome have a premature stop codon mutation in CLDN1, resulting in a failure of mature CLDN1 protein, and present ichthyosis, a skin disorder [10]. Therefore, CLDN1 must be necessary to maintain the physical barrier function in the skin.

The epidermis is exposed to ultraviolet (UV) radiation from the sun. UV is divided into three ranges on the basis of wavelength: UVA (320–400 nm), UVB (280–320 nm), and UVC (100–290 nm). Among them, UVC is blocked by the ozone layer, whereas UVA and UVB can penetrate the ozone layer and come into contact with the skin. UVB irradiation shows more cytotoxic and mutagenic effects than UVA irradiation in the experiments using human epidermis [11,12]. High dose of UVB (> 25 mJ/cm^2^) is usually used in the cytotoxicity experiments [13,14]. The UVB-induced cell death is caused by the generation of reactive oxygen species (ROS) including superoxide anion radical and hydroxy radical, or reactive nitrogen species (RNS) including nitric oxide (NO) and peroxynitrite [15]. The barrier function of the epidermis is attenuated by cell death. On the other hand, the effect of low dose of UVB on TJ barrier remains to be clarified.

NO is generated from L-arginine by NO synthases (NOS). NOS are divided into two classes, inducible (iNOS, NOS2) and constitutive NOS (cNOS). cNOS is further classified as neuronal NOS (nNOS, NOS1) and endothelial NOS (eNOS, NOS3). The protein expressions of NOS1-3 are detected in the human skin [16] and human keratinocyte-derived HaCaT cells [17,18]. UVB irradiation induces the elevation of NOS2 expression and phosphorylation of NOS3 in HaCaT cells. Both NO and peroxynitrite levels increase within 6 h post UVB irradiation. Peroxynitrite is a potent nitrating and oxidizing agent, leading to the nitration of tyrosine residue of target proteins and aberrations of their expression, localization, and function [19]. However, it is unknown what regulatory mechanism is involved in the elevation of NOS expression by UVB irradiation.

In the present study, we investigated the effect of low dose of UVB (5 mJ/cm^2^) on the expression and function of CLDN1 using HaCaT cells. In addition, the molecular mechanism of tight junctional localization of CLDN1 was assessed by immunoblotting, as well as immunofluorescence staining, RNS assay, and paracellular permeability measurements.

## 2. Results

### 2.1. Effect of UVB Irradiation on Cell Viability and the Cellular Localization of CLDN

The viability of HaCaT cells was examined by 2-(4-iodophenyl)-3-(4-nitrophenyl)-5-(2,4-disulfophenyl)-2H-tetrazolium (WST-1) assay. The exposure of the cells to weak UVB irradiation (5 mJ/cm^2^) induced a low toxicity, below 10% at 48 h (Figure 1A). CLDN1 was mainly localized in the cell–cell contact that is located at the apical region of lateral membranes, the so-called TJ, under control conditions (Figure 1B, Appendix A). The localization of CLDN1 was unchanged after 3 h of UVB irradiation, but it was significantly decreased after 6 h. Then, the tight junctional localization of CLDN1 was recovered after 48 h. In contrast, CLDN4 was diffusively distributed in the lateral membranes (Appendix A). The lateral localization of CLDN4 was unchanged by UVB irradiation (Figure 1C). In Western blotting, the protein level of CLDN1 was constant after UVB irradiation, whereas that of CLDN4 was transiently increased after 6 h and decreased after 48 h (Figure 1D). Similarly, the mRNA level of CLDN1 was constant after 6 h, whereas that of CLDN4 was increased (Figure 1E).

### 2.2. Effect of UVB Irradiation on Barrier Function

ZO-1 was mainly localized in the TJ, which was unchanged by UVB irradiation (Figure 2A). CLDN1 was colocalized with ZO-1 under control conditions (Appendix A). UVB irradiation induced the disappearance of CLDN1 in the TJ without affecting the localization of ZO-1. The tight junctional localization of CLDN1 was lost after 6 h of UVB irradiation. The total protein levels of CLDN1 and ZO-1 were unchanged after 6 h of UVB irradiation (Figure 2B and Appendix A). The TJ barrier was estimated by transepidermal electrical resistance (TER) and the flux of lucifer yellow (LY), a fluorescent hydrophilic marker. TER values were transiently decreased after 3–6 h of UVB irradiation, then they were returned to control level after 48 h (Figure 2C). Similarly, LY flux transiently increased after 6 h of UVB irradiation, and then it returned to control level after 48 h. These results indicate that weak UVB irradiation leads to the dissociation of CLDN1 from the TJ and the decrease in TJ barrier function.

### 2.3. Rescue of TJ Barrier Function by a Clathrin-Dependent Endocytosis Inhibitor

We recently reported that oxidative stress induces the mislocalization of CLDN1 in HaCaT cells, which is inhibited by monodansylcadaverine (MDC), a clathrin-dependent endocytosis inhibitor [20]. Similarly, the UVB-induced mislocalization of CLDN1 was inhibited by MDC, but not by methyl-β-cyclodextrin (MβCD), a caveolae-dependent endocytosis inhibitor (Figure 3A). The TJ barrier function including TER and LY flux was decreased by UVB irradiation, which was inhibited by MDC (Figure 3B). These results indicate that the clathrin-dependent endocytosis pathway may be involved in the decrease in CLDN1 localization at the TJ caused by weak UVB irradiation.

### 2.4. Increase in Tyrosine-Nitrated and Ubiquitinated CLDN1 by UVB

The cellular localization of CLDNs is controlled by various post-translational modifications such as phosphorylation, ubiquitination, nitration, and acetylation [21]. The phosphothreonine and phosphoserine levels of CLDN1 were unchanged by UVB irradiation (Figure 4A and Appendix A), which are different from the data of oxidative stress [20]. In contrast, the tyrosine nitration and mono-ubiquitination levels of CLDN1 were increased after 6 h of UVB irradiation (Figure 4B and Appendix A). The tyrosine nitration of target proteins is upregulated by RNS such as NO and peroxynitrite [22]. UVB irradiation increased NOS3 expression without affecting NOS2 expression (Figure 4C and Appendix A). NOS1 expression was not detected under our experimental conditions (data not shown). These results indicate that weak UVB irradiation induces the post-translational modifications of tyrosine-nitration and ubiquitination mediated by NOS3.

### 2.5. Involvement of RNS in UVB-Induced Mislocalization of CLDN1

Intracellular contents of NO, peroxynitrite, and ROS were monitored using 4,5-diaminofluorescein-2 diacetate (DAF-2DA), 1,3,5,7-tetramethyl-2,6-dicyano-4,4-difluoro-8-[2-(2-hydroxy-2-oxoethoxy)-4-hydroxyphenyl]-3a,4-dihydro-3a-aza-4a-azonia-4-bora(IV)-s-indacene (NiSPY-3), and 2′,7′-dichlorofluorescein diacetate (H_2_DCFDA), respectively. UVB irradiation increased both NO and peroxynitrite contents in a time-dependent manner and reached a peak after 3 h (Figure 5A). In contrast, the ROS content was not significantly changed. The UVB-induced elevation of NO and peroxynitrite contents was inhibited by *N*^G^-nitro-L-arginine methyl ester (L-NAME), a NOS inhibitor, and Fe(III)5,10,15,20-tetrakis(4-sulfonatophenyl)porphyrin (FeTPPs), a peroxynitrite scavenger (Figure 5B). 1-Hydroxy-2-oxo-3-(N-ethyl-2-aminoethyl)-3-ethyl-1-triazene (NOC12), a NO donor, increased NO content without exposure to UVB irradiation, whereas 3-(4-morpholinyl)sydnonimine hydrochloride (SIN-1), a peroxynitrite donor, did not. In contrast, peroxynitrite content was increased by both NOC12 and SIN-1. These results indicate that weak UVB irradiation, NOC12, and SIN-1 can increase the production of NO and peroxynitrite in HaCaT cells. Next, we investigated the effects of NO and peroxynitrite on the tight junctional localization of CLDN1. The UVB-induced mislocalization of CLDN1 was inhibited by L-NAME and FeTPPs (Figure 5C). In addition, both NOC12 and SIN-1 lead to the mislocalization of CLDN1 without exposure to UVB irradiation. The UVB-induced tyrosine-nitration of CLDN1 was inhibited by L-NAME (Appendix A). In contrast, NOC12 increased the tyrosine nitration of CLDN1 without UVB irradiation. These results indicate that the production of RNS may be involved in the tyrosine nitration and mislocalization of CLDN1 by UVB irradiation.

### 2.6. Destruction of TJ Barrier Function by RNS

We examined the effects of RNS on the TJ barrier function because RNS induced the mislocalization of CLDN1. NOC12 and SIN-1 decreased TER, and they increased paracellular LY flux (Figure 6). The UVB-induced decreases in TJ barrier function was rescued by L-NAME and FeTPPs. These results indicate that TJ barrier function is diminished by RNS. Therefore, we decided to investigate the mechanism of NO production by UVB irradiation.

### 2.7. Inhibition of UVB-Induced Responses by TRPV1 Antagonist

Transient receptor potential vanilloid (TRPV) is a molecular sensor for detecting adverse stimuli such as capsaicin, heat, and acid. TRPV1 and 4 are expressed in keratinocytes and function as a Ca^2+^ channel [23,24]. To clarify the Ca^2+^ influx pathway caused by UVB irradiation, we investigated the effects of TRPV1 and TRPV4 antagonists. The fluorescence intensity of Fluo-8, a Ca^2+^ indicator, was increased within 1 h after UVB irradiation (Figure 7A). The UVB-induced elevation of intracellular free Ca^2+^, NO, and peroxynitrite contents was inhibited by AMG9810, a TRPV1 antagonist, but not by RN1734, a TRPV4 antagonist (Figure 7B). Similarly, these UVB-induced responses were blocked by glycoletherdiaminetetraacetic acid (EGTA), a Ca^2+^ chelator. Olvanil, a TRPV1 agonist, increased intracellular free Ca^2+^, NO, and peroxynitrite contents without exposure to UVB irradiation (Figure 7C). These results indicate that TRPV1 may be involved in the elevation of Ca^2+^ influx and RNS production by UVB irradiation.

### 2.8. Inhibition of UVB-Induced Elevation of NOS3 and Mislocalization of CLDN1 by AMG9810

UVB irradiation increased NOS3 expression, which was inhibited by AMG9810 (Figure 8A and Appendix A). Olvanil increased NOS3 expression without exposure to UVB irradiation. These results are similar to those in intracellular free Ca^2+^, NO, and peroxynitrite contents. The mislocalization of CLDN1 caused by UVB irradiation was also inhibited by AMG9810 (Figure 8B). Olvanil caused the mislocalization of CLDN1 without UVB irradiation. These results indicate that TRPV1 may be involved in the UVB-induced elevation of NOS3 expression and mislocalization of CLDN1.

### 2.9. Inhibition of UVB-Induced Responses by Opsin 2 (OPN2) siRNA

Light is sensed by photoreceptors including OPN [25]. The expression of OPN subtypes was investigated by semi-quantitative polymerase chain reaction (PCR) assay. The bands of OPN2 and OPN3 were detected in HaCaT cells, whereas those of OPN1 and OPN5 were not (Figure 9A). The mRNA levels of OPN2 and OPN3 were decreased below 40% by each siRNA (Figure 9B). Fluorescence measurements indicated that OPN2 siRNA inhibited the UVB-induced elevation of intracellular free Ca^2+^, NO, and peroxynitrite contents (Figure 9C). In contrast, OPN3 siRNA had no inhibitory effects. The UVB-induced mislocalization of CLDN1 was rescued by OPN2 siRNA, not by OPN3 siRNA (Figure 9D). These results indicate that UVB may be sensed by OPN2 in HaCaT cells.

## 3. Discussion

UV exposure and oxidative stress may be involved in the disruption of the TJ barrier in the skin. In the previous research, TJ integrity was investigated using high doses of UV, which can induce noticeable cell damage [26]. The production of ROS is increased by UVB followed by decrease in cell viability. In the present study, we found that weak UVB irradiation induced the mislocalization of CLDN1 and destruction of TJ barrier function under non-toxic conditions (Figure 1 and Figure 2). Lower UVB (below 25 mJ/cm^2^) exposure to HaCaT cells has been reported to induce little damage, whereas high doses (above 50 mJ/cm^2^) cause apoptotic cell death [27]. We suggest that weak UVB irradiation can change construction and function of TJ independently of cell damage. The mRNA and protein levels of CLDN4 were increased by UVB irradiation, but UVB irradiation induced the reduction of TER and elevation of LY flux. The TER values were about 328.8 ± 40.6 (Ω∙cm2) in HaCaT cells, which indicate so-called leaky membrane. Therefore, we suggest that the transcellular pathway may contribute little to the reduction of TER. Both CLDN1 and ZO-1 were mainly distributed in the TJ under control conditions, whereas CLDN4 was diffusely distributed in the lateral membrane (Appendix A). CLDN4 may be insufficient to form integrated TJ in HaCaT cells. The other explanation is that the function of TJ is determined by homo- or heterophilic interactions of CLDNs [28]. CLDN4 may form TJ together with other CLDNs. After 48 h of weak UVB exposure, TER and LY flux were recovered to near the pre-exposed level in spite of the reduction of CLDN4 expression. Not only CLDN1 and CLDN4 mRNAs, but also CLDN7 and CLDN12 mRNAs were expressed in HaCaT cells (Appendix A). Other CLDNs without CLDN1 and CLDN4 may be also involved in the regulation of paracellular permeability.

Recently, El-Chami et al. [29] reported that a single dose of UVB (10 mJ/cm^2^) induces the elevation of ROS and disruption of the TJ barrier in rat epidermal keratinocyte. Our results indicated that the total amount of CLDN1 protein is constant for 24 h, but CLDN1 dissociates from the TJ (Figure 1). There is a discrepancy between our data and their report. We found that (i) low dose of UVB does not significantly elevate ROS production (Figure 5A), and (ii) CLDN1 disappears from the TJ, but CLDN4 does not. The discrepancy may be caused by the difference in experimental conditions: type of cell (human epidermal-derived HaCaT cells vs. rat epidermal cells), doses of UVB (5 mJ/cm^2^ vs. 10 mJ/cm^2^), and recovery periods (6 h vs. 24 h).

The human OPNs, which sense light, belong to the photosensitive G protein-coupled receptor superfamily that is divided into five subfamilies—OPN1-OPN5 [25]. OPN1-3 and 5 are reported to be expressed in the human epidermis [30]. OPN2 has a role in the regulation of the melanogenesis of melanocytes. OPN3 may contribute to hyperpigmentation induced by visible light in human melanocytes [31]. Human and mouse OPN5 has an absorption maximum at 380 nm and acts as a UV-sensitive photoreceptor to activate Gi-mediated light transduction pathways [32]. The mRNA expression of OPN2 and 3 was detected in HaCaT cells, whereas that of OPN5 was not (Figure 9A). The UVB-induced elevation of Ca^2+^, NO, and peroxynitrite contents was significantly inhibited by OPN2 siRNA, but not by OPN3 siRNA (Figure 9B–D). We suggest that OPN2, but not OPN3, can sense UVB in HaCaT cells. Similarly, UVA reduces mRNA levels of specific differentiation markers via upregulation of OPN2 in normal human epidermal keratinocytes, suggesting that OPN2 may serve as a sensor to UVA [33]. In addition, UVA induces immediate pigment darkening in murine melanocytes and melanoma cells [34]. However, the absorption spectra of OPN2 shows a peak at a longer wavelength of 505 nm, which is not in the UVA (320–400 nm) and UVB ranges (280–320 nm). There is a possibility that unknown sensor protein may function together with OPN2. Further study is needed to clarify how OPN2 is involved in the UVB-induced responses.

The activation of OPNs induces the opening of light-gated ion channels such as members of the TRP family [35]. A phosphoinositide cascade is suggested to be involved in the functional interaction between OPNs and TRP channel. Both TRPV1 and 4 are expressed in the skin [36]. UVB increased the fluorescence intensities of Fluo-8 (Ca^2+^ indicator), DAF-2 (NO indicator), and NiSPY-3 (peroxynitrite indicator) within several hours (Figure 5 and Figure 7). Especially, the fluorescence intensity of Fluo-8 was increased within 1 h. These results suggest that the intracellular free Ca^2+^ concentration is increased prior to the elevation of NO and peroxynitrite contents. The UVB-induced responses were inhibited by a TRPV1 antagonist (Figure 7 and Figure 8). Furthermore, olvanil, a TRPV1 agonist, induced the elevation of Ca^2+^, NO, and peroxynitrite contents and mislocalization of CLDN1. These results suggest that TRPV1 is involved in the weak UVB-induced response in HaCaT cells. UVB irradiation increases keratin 1 and 10 expression mediated via activation of TRPV1 in HaCaT cells [37]. The expression of NOS3 was increased by UVB irradiation, which was inhibited by TRPV1 antagonists (Figure 8A). Furthermore, olvanil, a TRPV1 agonist, increased the expression level of NOS3. The upregulation of NOS3 expression by calcium/calmodulin-dependent protein kinase II pathway is reported in endothelial cells [38]. The influx of Ca^2+^ through TRPV4 may be involved in the elevation of NOS3 expression in HaCaT cells. Further studies are needed to clarify what mechanism is involved in the UVB-induced elevation of NOS3 expression.

We recently reported that ROS production decreases the TJ localization of CLDN1 mediated by its dephosphorylation [20]. In contrast, weak UVB irradiation decreases the TJ localization of CLDN1 without affecting its phosphorylation level (Figure 4). The TJ localization of CLDN1 was rescued by MDC, suggesting that weak UVB irradiation enhances the clathrin-dependent endocytosis of CLDN1. However, we could not exclude the possibility that CLDN1 was widely diffused into the lateral membranes. De-phosphorylated CLDN1 by hypotonic stress dissociates from the TJ in Madin–Darby canine kidney cells [39]. Binge alcohol exposure increases nitrated CLDN1 level, leading to degradation via a ubiquitin-dependent proteolysis in human colonic T84 cells [40]. We found that weak UVB irradiation increases tyrosine nitration and ubiquitin levels of CLDN1 (Figure 4). The TJ localization of CLDN1 may be controlled by the post-translational modification including phosphorylation and nitration. The proposal scheme is depicted in Figure 10.

In conclusion, we found that weak UVB irradiation disrupted the TJ barrier function without affecting the viability of HaCaT cells. Weak UVB irradiation decreased the TJ localization of CLDN1, which was rescued by the inhibitors of NO and peroxynitrite production. The elevation of Ca^2+^, NO, and peroxynitrite contents was inhibited by OPN2 siRNA and TRPV1 inhibitor. We suggest that weak UVB can disrupt the TJ barrier mediated by OPN2/TRPV1/NOS3 signaling pathway in the skin.

## 4. Materials and Methods

### 4.1. Materials

Rabbit anti-CLDN1, mouse anti-CLDN4, rabbit anti-ZO-1, Alexa Fluor 488 anti-mouse, and Alexa Fluor 555 anti-rabbit antibodies, and H_2_DCFDA were obtained from Thermo Fisher Scientific (San Diego, CA, USA). Goat anti-β-actin antibody was from Santa Cruz Biotechnology (Santa Cruz, CA, USA). Mouse anti-phosphoserine (p-Ser) and anti-phosphothreonine (p-Thr) antibodies were from Sigma-Aldrich (Saint Louis, MO, USA). LY, NiSPY-3, Quest Fluo-8 AM, rabbit anti-nitrotyrosine, rabbit anti-NOS1, rabbit anti-NOS2, rabbit anti-NOS3, and rabbit anti-ubiquitine were from Biotium (Fremont, CA, USA), Goryo Kagaku (Hokkaido, Japan), AAT Bioquest (Sunnyvale, CA, USA), R&D Systems (Minneapolis, MN, USA), Cell Signaling Technology (Beverly, MA, USA), ProteinTech (Rosemont, IL, USA), GeneTex (Irvine, CA, USA), and Stressgen Biotechnologies Corporation (British Columbia, Canada), respectively. NOC12, SIN-1, and L-NAME were from Dojindo Laboratories (Kumamoto, Japan). AMG9810, olvanil, and RN1734 were from FUJIFILM Wako Pure Chemical Corporation (Tokyo, Japan). DAF-2DA and FeTPPs were from Cayman Chemical (Ann Arbor, MI, USA). All other reagents were of the highest grade of purity available.

### 4.2. Cell Culture

HaCaT cells, an immortalized non-tumorigenic human keratinocyte-derived cell line [41], were grown in Dulbecco’s modified Eagle’s medium (Sigma-Aldrich) supplemented with 5% fetal bovine serum (FBS; Sigma-Aldrich), 0.07 mg/mL penicillin-G potassium, and 0.14 mg/mL streptomycin sulfate in a 5% CO_2_ atmosphere at 37 °C. One day before experiments, we transferred cells to FBS-free medium.

### 4.3. UVB Irradiation

UVB irradiation was carried out using a UV Crosslinker CL-1000M (Analytik Jena, Upland, CA, USA), as described previously [20]. The cells were exposed to a total of 5 mJ/cm^2^ UVB at 0 h. UV exposure output was set by the equipment. Cell viability was examined using the WST-1 assay.

### 4.4. Confocal Microscopy

Cells cultured on cover glasses were incubated with rabbit anti-CLDN1, mouse anti-CLDN4, or rabbit anti-ZO-1 antibodies (1:50 dilution). The fluorescence images were taken near the apical membrane using an LSM 700 confocal microscope (Carl Zeiss, Jena, Germany), as described previously [20].

### 4.5. Paracellular Permeability

Cells were cultured on Transwell plates (0.4 μm pore size, Corning Inc., Corning, NY, USA). TER and the paracellular permeability to LY, a fluorescent paracellular flux marker, were measured as described previously [20].

### 4.6. SDS-Polyacrylamide Gel Electrophoresis and Immunoblotting

In immunoprecipitation assay, we incubated the cell lysates with anti-CLDN1 antibody and protein G-Sepharose beads for 16 h at 4 °C with gentle rocking. SDS-polyacrylamide gel electrophoresis (SDS-PAGE) and immunoblotting were performed as described previously [20]. Band density was quantified using ImageJ software (National Institute of Health software). The signals were normalized using the β-actin loading control. In immunoprecipitation assay, we compensated the signals by CLDN1.

### 4.7. Intracellular RNS and Ca^2+^ Contents

HaCaT cells were incubated with DAF-2DA, a fluorescent NO indicator; NiSPY-3, a fluorescent peroxynitrite indicator; and Fluo-8 AM, a fluorescent Ca^2+^ indicator, for 30 min. The fluorescence intensities of these indicators were detected using an Infinite F200 Pro microplate reader.

### 4.8. Isolation of Total RNA and PCR

Total RNA was extracted using TRI reagent (Sigma-Aldrich). Reverse transcription was carried out using ReverTraAce (Toyobo Life Science, Osaka, Japan). In the case of real-time PCR, the reaction was performed as described previously [42] using the primer pairs for CLDNs (Table 1 and Appendix A). In the case of semi-quantitative PCR, the reaction was performed using the primer pairs for OPN1, 2, 3, and 5 (Table 2). The PCR products were visualized after electrophoresis on an agarose gel.

### 4.9. Transfection of Plasmid DNA and siRNA

siRNAs against OPN2 (SASI_Hs01_00065400), OPN3 (SASI_Hs01_00345429), and a negative control (SIC-001) were purchased from Sigma-Aldrich. The siRNAs were transfected into cells using Screen FectA (FUJIFILM Wako Pure Chemical Corporation) as recommended by the manufacturer.

### 4.10. Statistical Analysis

The data are presented as means ± standard error of the mean. Differences between groups were analyzed using one-way analysis of variance, and corrections for multiple comparison were made using Tukey’s multiple comparison test. Comparisons between two groups were made using Student’s *t*-test. Statistical analyses were performed using KaleidaGraph version 4.5.1 software (Synergy Software, Reading, PA, USA). Significant differences were assumed at *p* < 0.05.

## Figures and Tables

**Figure 1 ijms-21-07138-f001:**
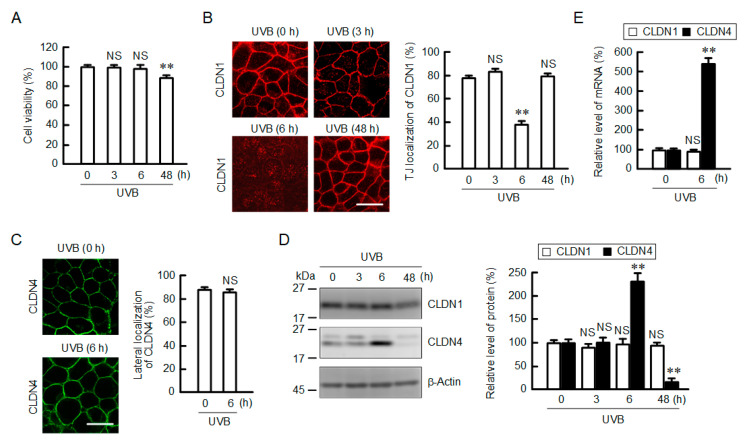
Effect of weak ultraviolet B (UVB) irradiation on cell viability and localization of claudin (CLDN). (**A**) After exposing to weak UVB (5 mJ/cm^2^), the cells were cultured for 0–48 h. Cell viability was measured using a 2-(4-iodophenyl)-3-(4-nitrophenyl)-5-(2,4-disulfophenyl)-2H-tetrazolium (WST-1) assay. *n* = 6. (**B**,**C**) Cells exposed to UVB were cultured for 0–48 h. The cells were immunostained with anti-CLDN1 antibody (**B**) and anti-CLDN4 antibodies (**C**). Scale bars indicate 10 μm. The fluorescence values of CLDN1 at the tight junction (TJ) and CLDN4 at the lateral side are shown as a percentage of the total fluorescence values. *n* = 3. (**D**) Cell lysates were applied to 12.5% SDS-PAGE followed by blotting with anti-CLDN1, anti-CLDN4, and anti-β-actin antibodies. The full-length blot images are shown in Appendix A. The levels of CLDN1 and CLDN4 are represented as a percentage of 0 h. *n* = 4. (**E**) The expression levels of CLDN1 and CLDN4 mRNAs were measured using real-time PCR. *n* = 4. ** *p* < 0.01 and not significant (NS) *p* > 0.05 vs. 0 h.

**Figure 2 ijms-21-07138-f002:**
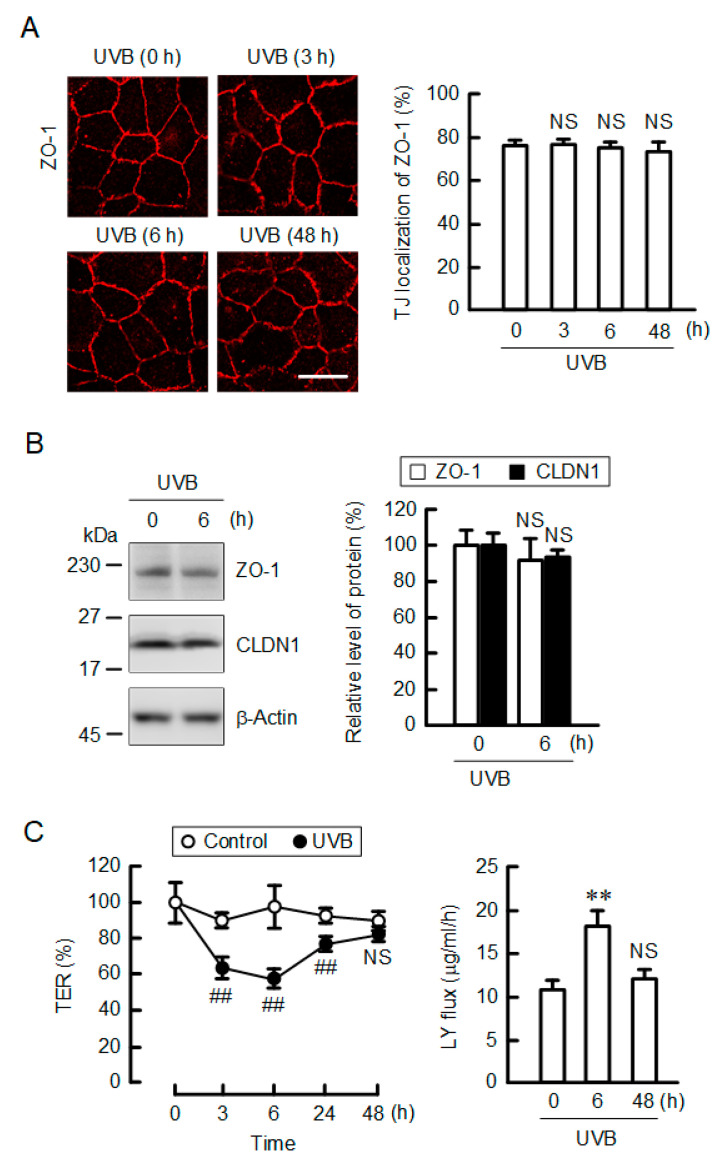
Effect of weak UVB irradiation on TJ barrier function. (**A**) After exposing to weak UVB, the cells were cultured for indicated periods. The cells were immunostained with anti-zonula occludens-1 (ZO-1) antibody. Scale bars indicate 10 μm. The fluorescence values of ZO-1 at the TJ are shown as a percentage of the total fluorescence values. *n* = 4. (**B**) Cell lysates were applied to SDS-PAGE followed by blotting with anti-ZO-1, anti-CLDN1, and anti-β-actin antibodies. The full-length blot images are shown in Appendix A. The expression levels of ZO-1 and CLDN1 are represented as a percentage of 0 h. *n* = 3. (**C**) After exposing to weak UVB, the cells were cultured for 0–48 h. Control cells were cultured without exposure to UVB. Transepidermal electrical resistance (TER) was measured using a volt-ohmmeter and represented as a percentage of the value at 0 h. Paracellular lucifer yellow (LY) flux was analyzed using a fluorescence spectrometry. *n* = 6. ** *p* < 0.01 vs. 0 h. ^##^
*p* < 0.01 vs. control cells. NS *p* > 0.05 vs. 0 h or control cells.

**Figure 3 ijms-21-07138-f003:**
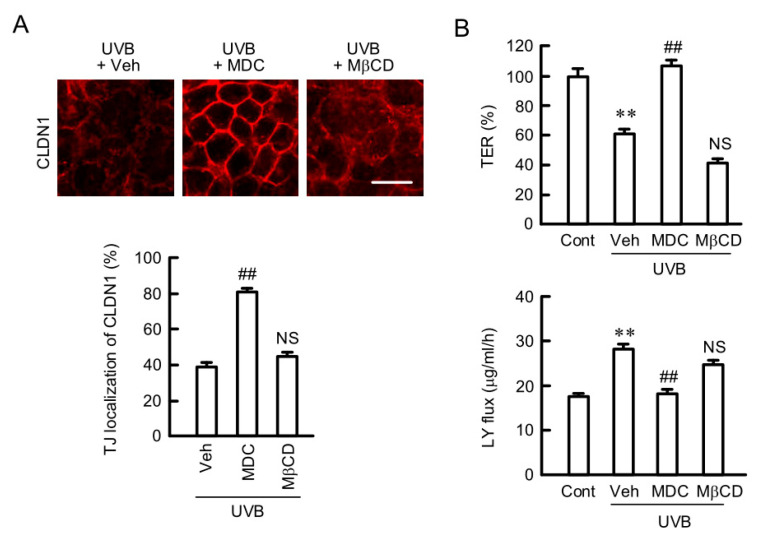
Effects of endocytosis inhibitors on the destruction of TJ barrier by weak UVB irradiation. After exposing to weak UVB, the cells were cultured for 6 h in the absence (Veh) and presence of 5 μM monodansylcadaverine (MDC) or 10 μM methyl-β-cyclodextrin (MβCD). Control cells (Cont) were cultured without exposure to UVB. (**A**) The cells were immunostained with anti-CLDN1 antibody. Scale bar indicates 10 μm. The fluorescence values of CLDN1 at the TJ are shown as a percentage of the total fluorescence values. *n* = 4. (**B**) TER was measured using a volt-ohmmeter and represented as a percentage of the value of control. Paracellular LY flux was analyzed using a fluorescence spectrometry. *n* = 6. ** *p* < 0.01 vs. Cont. ^##^
*p* < 0.01 and NS *p* > 0.05 vs. Veh.

**Figure 4 ijms-21-07138-f004:**
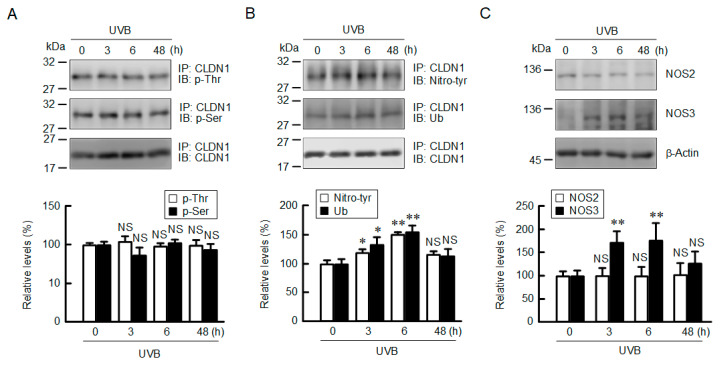
Effect of weak UVB irradiation on phosphorylation, ubiquitination, and tyrosine nitration of CLDN1. After exposing to weak UVB, the cells were cultured for the period indicated. (**A**,**B**) After we performed immunoprecipitation with anti-CLDN1 antibodies, the immunoprecipitants were applied to SDS-PAGE and blotted with anti-phosphothreonine (p-Thr), anti-phosphoserine (p-Ser), anti-tyrosine nitration (Nitro-tyr), anti-ubiquitin (Ub), and anti-CLDN1 antibodies. The full-length blot images are shown in Appendix A. The levels of p-Thr, p-Ser, Nitro-tyr, and Ub of CLDN1 are represented as a percentage of 0 h. *n* = 3. (**C**) Cell lysates were applied to SDS-PAGE followed by blotting with anti-NO synthase 2 (NOS2), anti-NOS3, and anti-β-actin antibodies. The levels of NOS2 and NOS3 are represented as a percentage of 0 h. *n* = 4. ** *p* < 0.01, * *p* < 0.05, and NS *p* > 0.05 vs. 0 h.

**Figure 5 ijms-21-07138-f005:**
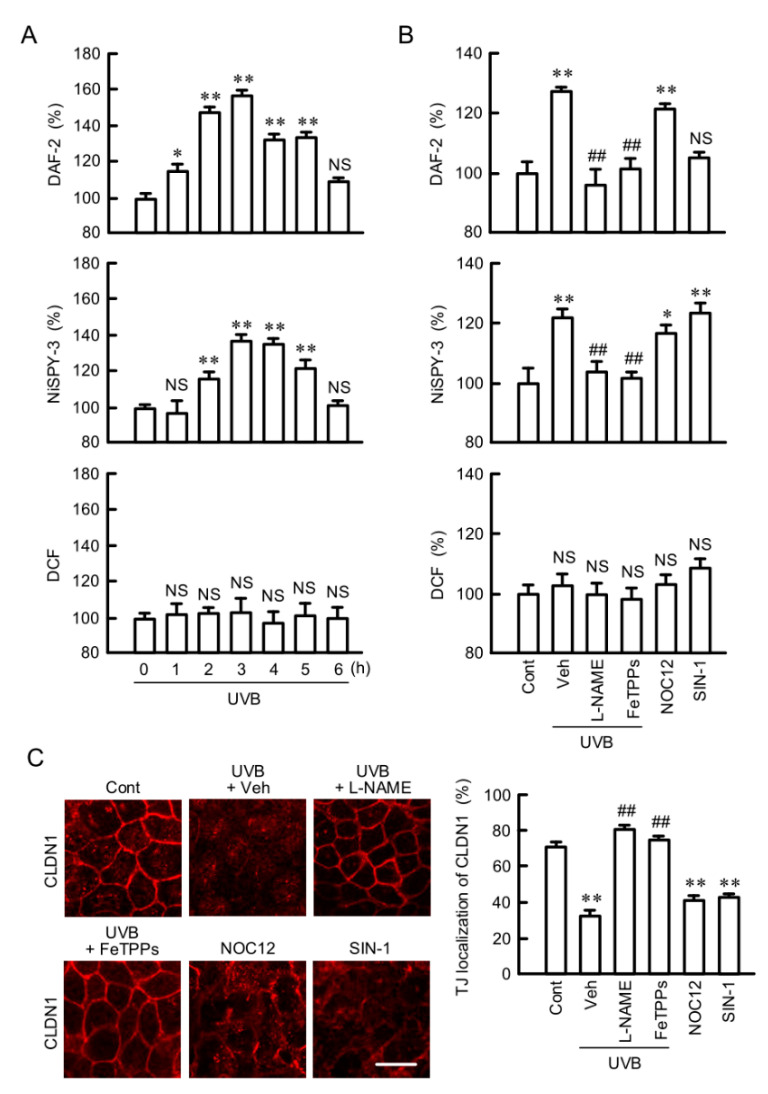
Involvement of reactive nitrogen species (RNS) in the UVB-induced mislocalization of CLDN1. (**A**) After exposing to weak UVB, the cells were cultured for between 0 and 6 h. The cells were incubated with 5 μM 4,5-diaminofluorescein-2 diacetate (DAF-2DA), 5 μM 1,3,5,7-tetramethyl-2,6-dicyano-4,4-difluoro-8-[2-(2-hydroxy-2-oxoethoxy)-4-hydroxyphenyl]-3a,4-dihydro-3a-aza-4a-azonia-4-bora(IV)-s-indacene (NiSPY-3), or 5 μM 2′,7′-dichlorofluorescein diacetate (H_2_DCFDA) for 30 min. The fluorescence intensities of DAF-2, NiSPY-3, and DCF were measured using a plate reader. (**B**,**C**) After exposing to weak UVB, the cells were cultured for 3 h (**B**) or 6 h (**C**) in the absence (Veh) and presence of 100 μM *N*^G^-nitro-L-arginine methyl ester (L-NAME) or 5 μM Fe(III)5,10,15,20-tetrakis(4-sulfonatophenyl)porphyrin (FeTPPs). In the case of UVB-untreated samples, cells were incubated in the absence (Cont) and presence of 10 μM 1-hydroxy-2-oxo-3-(N-ethyl-2-aminoethyl)-3-ethyl-1-triazene (NOC12) or 250 μM 3-(4-morpholinyl)sydnonimine hydrochloride (SIN-1). The fluorescence intensities of DAF-2, NiSPY-3, and DCF were measured using a plate reader (**B**). *n* = 8. The cells were immunostained with anti-CLDN1 antibody (**C**). Scale bar indicates 10 μm. The fluorescence values of CLDN1 at the TJ are shown as a percentage of the total fluorescence values. *n* = 4. ** *p* < 0.01, * *p* < 0.05, and NS *p* > 0.05 vs. 0 h, Cont, or Veh. ^##^
*p* < 0.01 vs. Veh.

**Figure 6 ijms-21-07138-f006:**
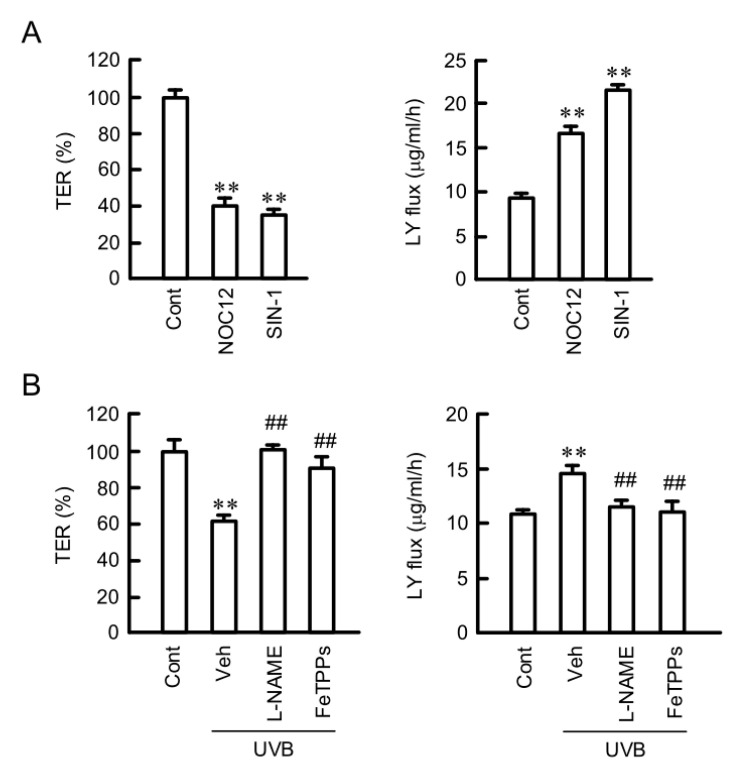
Involvement of RNS in the UVB-induced destruction of TJ. (**A**) Cells were incubated for 6 h in the absence (Cont) and presence of 10 μM NOC12 or 250 μM SIN-1. (**B**) After exposing to weak UVB, the cells were incubated for 6 h in the absence (Veh) and presence of 100 μM L-NAME or 5 μM FeTPPs. TER was measured using a volt-ohmmeter and represented as a percentage of the value of control (Cont). Paracellular LY flux was analyzed using a fluorescence spectrometry. *n* = 6. ** *p* < 0.01 vs. Cont. ^##^
*p* < 0.01 vs. Veh.

**Figure 7 ijms-21-07138-f007:**
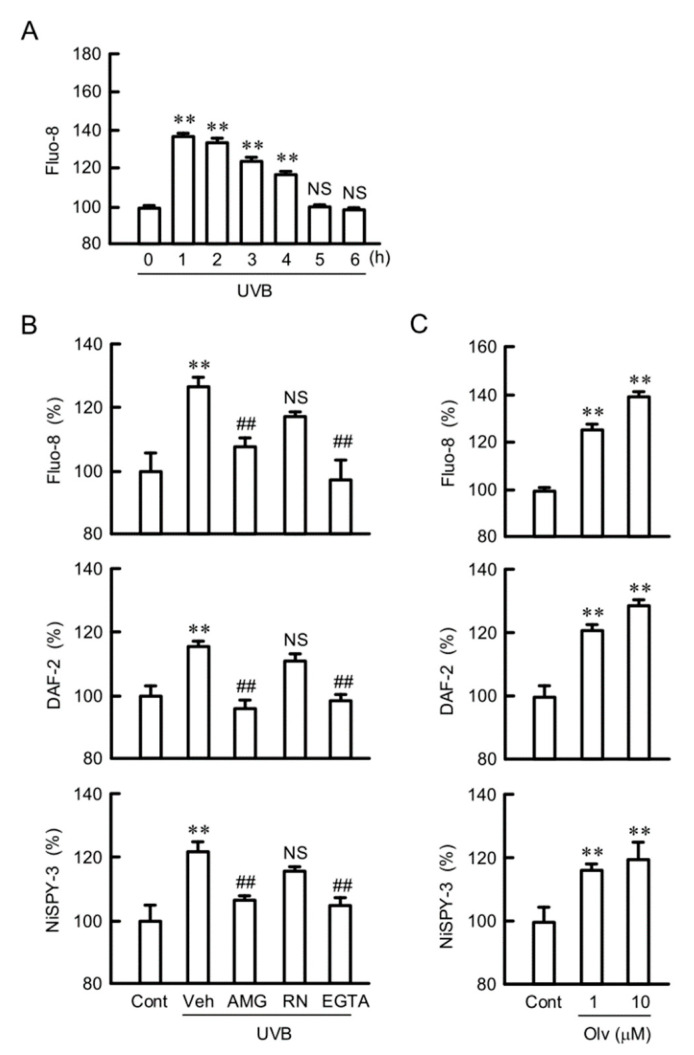
Involvement of transient receptor potential type vanilloid 1 (TRPV1) in the UVB-induced production of RNS. (**A,B**) After we exposed cells to weak UVB, the cells were incubated for 3 h in the absence (Veh) and presence of 10 μM AMG9810 (AMG), 10 μM RN1734 (RN), or 2 mM glycoletherdiaminetetraacetic acid (EGTA). (**C**) Cells were incubated for 3 h in the absence (Cont) and presence of olvanil (Olv). The cells were incubated with 5 μM Fluo-8 AM, 5 μM DAF-2DA, or 5 μM NiSPY-3 for 30 min. The fluorescence intensities of Fluo-8, DAF-2, and NiSPY-3 were measured using a plate reader. *n* = 8. ** *p* < 0.01 vs. Cont. ^##^
*p* < 0.01 and NS *p* > 0.05 vs. Veh.

**Figure 8 ijms-21-07138-f008:**
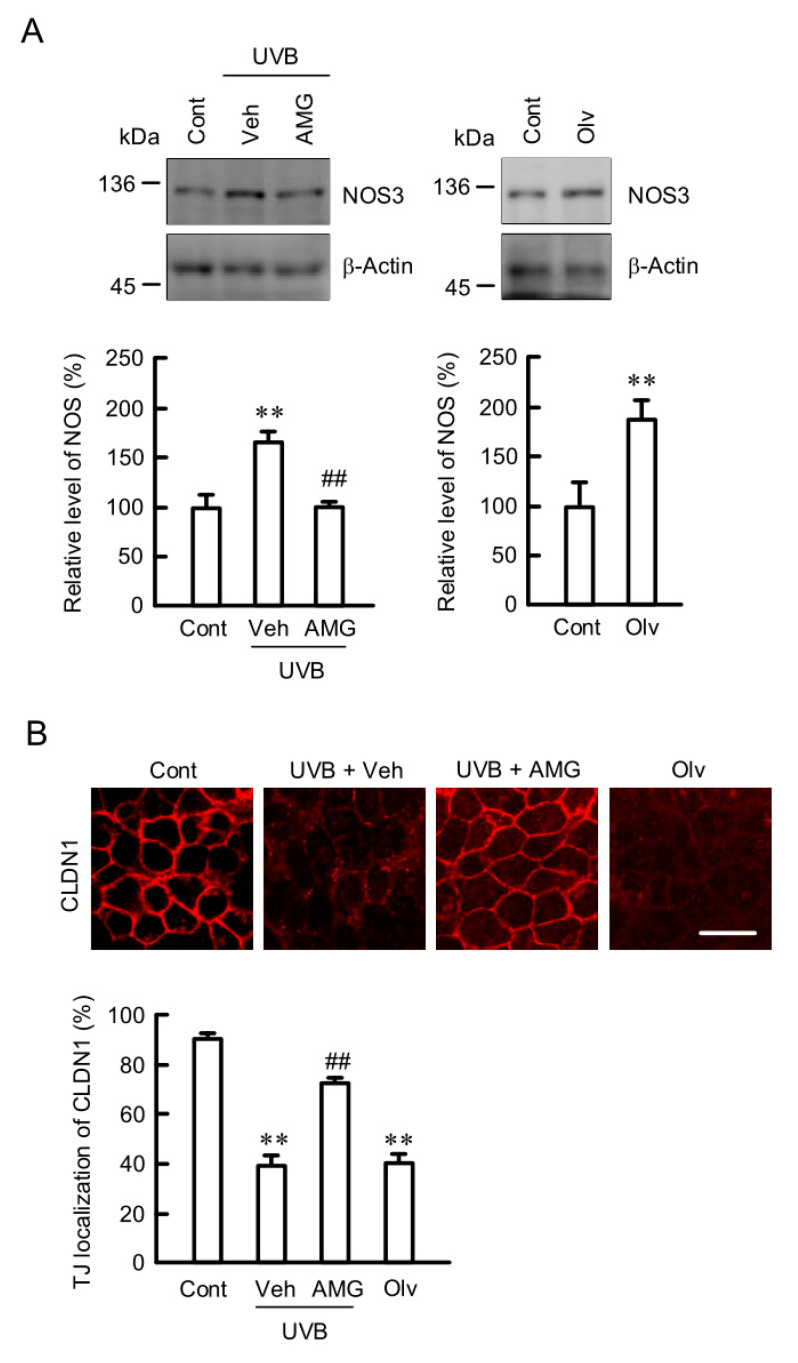
Involvement of TRPV1 in the UVB-induced destruction of TJ. (**A**) After exposing to weak UVB, the cells were incubated for 6 h in the absence (Veh) and presence of 10 μM AMG9810 (AMG). In the case of UVB-untreated samples, cells were incubated for 6 h in the absence (Cont) and presence of 1 μM olvanil (Olv). The cell lysates were applied to 7.5% SDS-PAGE followed by blotting with anti-NOS3 and anti-β-actin antibodies. The full-length blot images are shown in Appendix A. The level of NOS3 is represented as a percentage Cont. *n* = 4. (**B**) The cells were immunostained with anti-CLDN1 antibody. Scale bar indicates 10 μm. The fluorescence values of CLDN1 at the TJ are shown as a percentage of the total fluorescence values. *n* = 6. ** *p* < 0.01 and NS *p* > 0.05 vs. Cont. ^##^
*p* < 0.01 vs. Veh.

**Figure 9 ijms-21-07138-f009:**
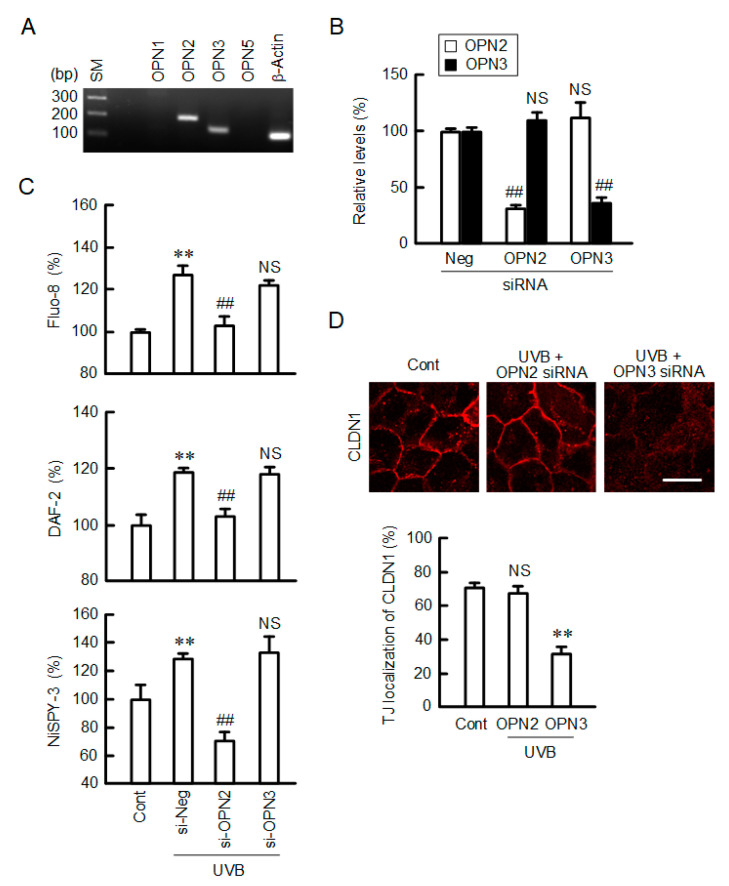
Involvement of OPN2 in the UVB-induced production of RNS. (**A**) PCR was performed using primer pairs against OPN1-3 and 5, as well as β-actin. The PCR products were visualized after electrophoresis on 2% agarose gel. Size marker (SM) was loaded on the left lane. *n* = 3. (**B**) The cells were transfected with negative (si-Neg), OPN2, or OPN3 siRNA and cultured for 3 days. The expression levels of OPN2 and OPN3 mRNAs were measured using real-time PCR. *n* = 4. (**C**) The cells transfected with si-Neg, OPN2, or OPN3 siRNA were exposed to weak UVB. After 3 h, the cells were incubated with 5 μM Fluo-8 AM, 5 μM DAF-2DA, or 5 μM NiSPY-3 for 30 min. The fluorescence intensities of Fluo-8, DAF-2, and NiSPY-3 were measured using a plate reader. *n* = 6. (**D**) The cells transfected with si-Neg, OPN2, or OPN3 siRNA were exposed to weak UVB. The cells were immunostained with anti-CLDN1 antibody. Scale bar indicates 10 μm. The fluorescence values of CLDN1 at the TJ are shown as a percentage of the total fluorescence values. *n* = 5. ** *p* < 0.01 vs. Cont. ^##^
*p* < 0.01 and NS *p* > 0.05 vs. si-Neg.

**Figure 10 ijms-21-07138-f010:**
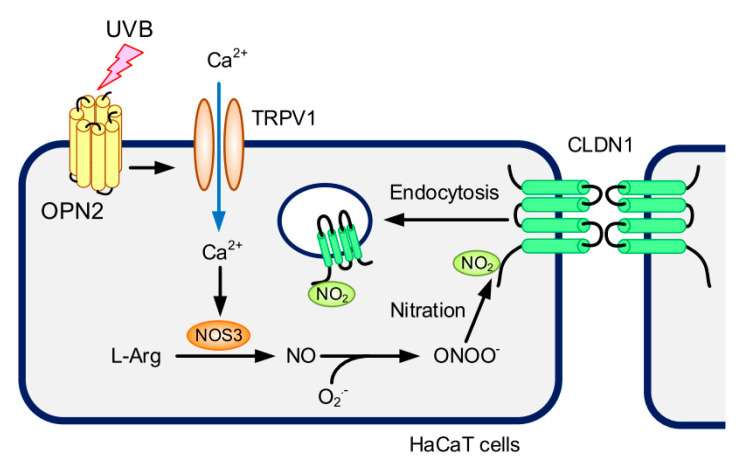
A proposal scheme for UVB responses in HaCaT cells.

**Table 1 ijms-21-07138-t001:** Primer pairs for real-time PCR.

Name	Direction	Sequence
CLDN1	Forward	5′-ATGAGGATGGCTGTCATTGG-3′
	Reverse	5′-ATTGACTGGGGTCATAGGGT-3′
CLDN4	Forward	5′-TTGTCACCTCGCAGACCATC-3′
	Reverse	5′-GCAGCGAGTCGTACACCTTG-3′
β-Actin	Forward	5′-CCTGAGGCACTCTTCCAGCCTT-3′
	Reverse	5′-TGCGGATGTCCACGTCACACTTC-3′

**Table 2 ijms-21-07138-t002:** Primer pairs for semi-quantitative PCR.

Name	Direction	Sequence
OPN1	Forward	5′-CTGCATCTTCTCTGTCTTCCCT-3′
	Reverse	5′-CAGTGACCATCCTGTAACCAGA-3′
OPN2	Forward	5′-GCACAGAAGGCCCTAACTTCTA-3′
	Reverse	5′-CGTAGAGCGTGAGGAAGTTGAT-3′
OPN3	Forward	5′-TGGTGCTCGTCCTCTACTACAA-3′
	Reverse	5′-AGGACACGAAGGTAAAGGTGAC-3′
OPN5	Forward	5′-ACTCCATTCCCATACAGCTCTC-3′
	Reverse	5′-ACTTCCTGACTGTGGTTACGGT-3′
β-Actin	Forward	5′-CCTGAGGCACTCTTCCAGCCTT-3′
	Reverse	5′-TGCGGATGTCCACGTCACACTTC-3′

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
