# Peer review of "Weak Ultraviolet B Enhances the Mislocalization of Claudin-1 Mediated by Nitric Oxide and Peroxynitrite Production in Human Keratinocyte-Derived HaCaT Cells"

_ijms, 2020, doi:10.3390/ijms21197138_

Round 1

Reviewer 1 Report

The paper entitled “Weak Ultraviolet B Enhances the Mislocalization of Claudin-1 Mediated by Nitric Oxide and Peroxynitrite Production in Human Keratinocyte-Derived HaCaT Cells” by Kobayashi et al. describes the signaling pathway involved in the TJ barrier disruption induced by UVB. However, there is no clear explanation about the purpose of this study such as what physiological or pathological situation is supposed in this study and what is the significance of the results. In addition, there are several aspects to address before publication as listed below.

Major points

The authors claim that claudin-1 localization is changed by UVB and the change is correlated to the disruption of TJ barrier function. However, claudin localization is not restricted at TJs but they localize at entire lateral cell membranes. The authors should evaluate the localization of claudin-1 at TJs by staining with TJ markers such as ZO-1 and showing z axis image.

The authors examine the effects of UVB on claudin-1 and claudin-4 only and conclude that the changes in TER and lucifer flux by UVB are contributed by claudin-1. Does HaCaT cell express only these two claudins?

The authors claim that UVB affects a TJ barrier mainly based on the changes in TER. However, in the epithelia with high TER (so called ‘tight’ epithelia), the TER is also affected by a transcellular pathway. What is the absolute value of TER in HaCaT cells? The authors should show it and discuss the effect of the transcellular pathway in TER.

Minor points

In figure 1, claudin-1 staining is reduced but the protein amount is not changed. Also, the protein amount of claudin-4 is obviously increased but the staining is not changed. The authors should mention the reason of the discrepancy in the staining and the blotting.

In figure 4, the authors claim that tyrosine nitration is increased 6 hours after UVB irradiation and the NOS3 expression returns to the original level 48 hours after UVB irradiation, but the results of blotting are not clear. The authors should describe the reason of the discrepancy in the blotting and quantification data.

It is claimed that NOS increases NO, which results in tyrosine nitration in claudin-1. However, the tyrosine nitration of claudin-1 in the presence of a NOS inhibitor and a NO donor is not shown. The results which demonstrate the authors claim should be included in the manuscript.

The role of opsins is examined using siRNA in figure 9 but there is no data about protein expression levels of opsins after siRNA treatment. The levels of suppression by the siRNA should be shown in the results.

In figure 10, the authors claim that opsin-2 acts as UVB sensor and induces claudin-1 endocytosis, but the effect of opsin-2 siRNA on claudin-1 localization is not shown. The authors should show the evidence of the scheme in figure 10.

As discussed by the authors, an absorption spectrum of opsin-2 is different from that of UVB. However, the authors claim that opsin-2 acts as a sensor of UVB. The reason of the authors idea and mechanism should be discussed in more detail.

The paragraph of lines 288-305 is mainly a summary of the results and includes little discussion. The authors should revise it.

Author Response

We thank you very much for your careful reading of our manuscript and valuable comments.

Comment 1

The authors claim that claudin-1 localization is changed by UVB and the change is correlated to the disruption of TJ barrier function. However, claudin localization is not restricted at TJs but they localize at entire lateral cell membranes. The authors should evaluate the localization of claudin-1 at TJs by staining with TJ markers such as ZO-1 and showing z axis image.

Answer

  Following your suggestion, we performed additional experiments. Both claudin-1 and ZO-1 were distributed at the apical side of lateral membrane, whereas claudin-4 was diffusely distributed in the lateral membrane. Please see new supplementary figure S7. We need further study to clarify the interaction between claudin-1 and ZO-1, and the colocalization of claudin-1 with ZO-1.

Comment 2

The authors examine the effects of UVB on claudin-1 and claudin-4 only and conclude that the changes in TER and lucifer flux by UVB are contributed by claudin-1. Does HaCaT cell express only these two claudins?

Answer

  Following your suggestion, we performed additional experiments. We found that not only claudin-1 and claudin-4, but also claudin-7 and claudin-12 are expressed in HaCaT cells. Please see new supplementary figure S9. We discussed the involvement of these claudins. Please see line 279.

Comment 3

The authors claim that UVB affects a TJ barrier mainly based on the changes in TER. However, in the epithelia with high TER (so called ‘tight’ epithelia), the TER is also affected by a transcellular pathway. What is the absolute value of TER in HaCaT cells? The authors should show it and discuss the effect of the transcellular pathway in TER.

Answer

  Following your suggestion, we showed the value of TER. Please see line 274. In addition, we discussed the effect of transcellular pathway on TER. Please see line 275.

Comment 4

In figure 1, claudin-1 staining is reduced but the protein amount is not changed. Also, the protein amount of claudin-4 is obviously increased but the staining is not changed. The authors should mention the reason of the discrepancy in the staining and the blotting.

Answer

  As your indication, claudin-1 staining was reduced but the protein amount was not changed after 6 h of UVB exposure. We suggest that claudin-1 was diffused into the cytosol. The protein amount of claudin-4 is obviously increased but the staining is not changed. As shown in supplementary figure S7, claudin-4 was diffusively distributed in the lateral membrane. Therefore, we may not be able to detect the change of staining.

Comment 5

In figure 4, the authors claim that tyrosine nitration is increased 6 hours after UVB irradiation and the NOS3 expression returns to the original level 48 hours after UVB irradiation, but the results of blotting are not clear. The authors should describe the reason of the discrepancy in the blotting and quantification data.

Answer

  We quantified the levels of tyrosine nitration of claudin-1 and NOS3 expression using the data of four independent experiments. We think the data of blotting are matched with the data of quantification.

Comment 6

It is claimed that NOS increases NO, which results in tyrosine nitration in claudin-1. However, the tyrosine nitration of claudin-1 in the presence of a NOS inhibitor and a NO donor is not shown. The results which demonstrate the authors claim should be included in the manuscript.

Answer

  Following your suggestion, we performed additional experiments. The tyrosine nitration of claudin-1 was decreased by a NOS inhibitor and increased by a NO donor. Please see new supplementary figure S8.

Comment 7

The role of opsins is examined using siRNA in figure 9 but there is no data about protein expression levels of opsins after siRNA treatment. The levels of suppression by the siRNA should be shown in the results.

Answer

  Following your suggestion, we performed additional experiments. We could not obtain specific antibodies against opsin2 and opsin3. Therefore, we examined the effects of siRNAs on mRNA expression of opsins. Please see new figure 9B.

Comment 8

In figure 10, the authors claim that opsin-2 acts as UVB sensor and induces claudin-1 endocytosis, but the effect of opsin-2 siRNA on claudin-1 localization is not shown. The authors should show the evidence of the scheme in figure 10.

Answer

  Following your suggestion, we performed additional experiments. We found that the UVB-induced mislocalization of claudin-1 was rescued by opsin2 siRNA. Please see new figure 9D.

Comment 9

As discussed by the authors, an absorption spectrum of opsin-2 is different from that of UVB. However, the authors claim that opsin-2 acts as a sensor of UVB. The reason of the authors idea and mechanism should be discussed in more detail.

Answer

  Following your suggestion, we discussed the involvement of other sensor protein. Please see line 307.

Comment 10

The paragraph of lines 288-305 is mainly a summary of the results and includes little discussion. The authors should revise it.

Answer

  Following your suggestion, we rewrote the section.

Reviewer 2 Report

In the study of Kobayashi et al. the effect of low UVB radiation on the paracellular skin barrier is analyzed focusing on claudin-1. The authors observe dislocation of claudin-1 within 6 hours, which is reversible after 48 h. They further analyze the regulatory processes behind this and finally link the photosensitive receptor OPN2 to this, with a subsequent activation of TRPV1 channels and induction of NO and peroxynitrite production stimulating endocytosis of claudin-1.

Although the paper is treating a very interesting topic and the presented data are convincing, there are a few points that should be addressed with care.

Major:

  • Counterstainings with an unaffected TJ-marker should be used to show the dislocation from TJs in all the staining images shown. Especially for the claudin-4 image (Fig. 1 C) it is difficult to say whether the cells are out of the level shown at 0 h and thus there appears to be weak claudin-4 signals or if there is an increase after 6 h. Best would be to show z-stacks to evaluate whether there maybe is also shift into more basolateral regions for example.
  • The duration of UVB treatment should be clarified. In the manuscript it reads as there was a 3, 6, 48 h incubation with UVB, while in other parts it is mentioned as recovery of 6 h. The methods section gives no clear hint.
    In this context, would one not expect an accumulating effect when the cells are exposed to the low does over time?
  • How is the cell viability after 48 h? It is confusing that cell viability is shown for 24 h and all other “recovery effects” are at 48 h? How are claudin-1 and claudin-4 after 24 h?
  • It is interesting that there appears to be a huge effect on claudin-4. The authors already reported similar effects in their recent publication but did not take them in further account. There is a huge upregulation of claudin-4 after 6 h. If the staining shown in Fig. 1C suggests due to missing counterstaining a localization of this double amount of claudin-4 in the TJ, would one not expect to see functional changes at least in TER as claudin-4 is described a mainly barrier-forming TJ protein?
    In addition, there seems to be no real recovery after 48 h because claudin-4 is not back to normal levels but nearly absent. This for example the authors did not see in their recent paper, which makes claudin-4 even more interesting. It is surprising that the TER is back to the values of t0 when claudin-4 is downregulated in such an extent. Is there a counter-regulation by another claudin? Maybe the authors should think about discriminating between a short term effect as they observe it for claudin-1 and long term effect that may indicate that there are further changes, for example for claudin-4 and presumably others if the properties of the barrier seem to be restored.
  • The authors state that ZO-1 is unchanged In their experiments. However, they only show 0 and 6 h. All the timepoints should be presented as shown for claudin-1 and -4.

Minor:

  • The authors should mention in the discussion that the 5 mJ/cm² dose of UVB does not cause an increase of ROS, as they show in their recent paper. It would make it easier to follow without having to look it up.
  • The text needs some rephrasing and clearer formulation, especially in the introduction.

Author Response

We thank you very much for your careful reading of our manuscript and valuable comments.

Comment 1

Counter stainings with an unaffected TJ-marker should be used to show the dislocation from TJs in all the staining images shown. Especially for the claudin-4 image (Fig. 1 C) it is difficult to say whether the cells are out of the level shown at 0 h and thus there appears to be weak claudin-4 signals or if there is an increase after 6 h. Best would be to show z-stacks to evaluate whether there maybe is also shift into more basolateral regions for example.

Answer

Following your suggestion, we investigated the localization of claudin-4. The xz images showed that claudin-1 and ZO-1 were localized at the apical side in the lateral membrane, whereas claudin-4 was diffusively distributed in the lateral membrane. Please see new supplementary figure S7.

Comment 2

The duration of UVB treatment should be clarified. In the manuscript it reads as there was a 3, 6, 48 h incubation with UVB, while in other parts it is mentioned as recovery of 6 h. The methods section gives no clear hint. In this context, would one not expect an accumulating effect when the cells are exposed to the low does over time?

Answer

Following your suggestion, we described the detail of UVB treatment. Please see line 366.

Comment 3

How is the cell viability after 48 h? It is confusing that cell viability is shown for 24 h and all other “recovery effects” are at 48 h? How are claudin-1 and claudin-4 after 24 h?

Answer

  We showed the cell viability after 48 h of UVB exposure. Please see new figure 1A.

Comment 4

It is interesting that there appears to be a huge effect on claudin-4. The authors already reported similar effects in their recent publication but did not take them in further account. There is a huge upregulation of claudin-4 after 6 h. If the staining shown in Fig. 1C suggests due to missing counterstaining a localization of this double amount of claudin-4 in the TJ, would one not expect to see functional changes at least in TER as claudin-4 is described a mainly barrier-forming TJ protein?

Answer

  Both claudin-1 and ZO-1 were mainly distributed in the apical side of lateral membrane, whereas claudin-4 was diffusely distributed in the lateral membrane (Supplementary figure S7). Claudin-4 may be insufficient to form integrated TJ in HaCaT cells. The other explanation is that the function of TJ is determined by homo- or heterophilic interactions of claudins. Claudin-4 may not form integrated tight junction alone. We discussed it in the Discussion. Please see line 276.

Comment 5

In addition, there seems to be no real recovery after 48 h because claudin-4 is not back to normal levels but nearly absent. This for example the authors did not see in their recent paper, which makes claudin-4 even more interesting. It is surprising that the TER is back to the values of t0 when claudin-4 is downregulated in such an extent. Is there a counter-regulation by another claudin? Maybe the authors should think about discriminating between a short term effect as they observe it for claudin-1 and long term effect that may indicate that there are further changes, for example for claudin-4 and presumably others if the properties of the barrier seem to be restored.

Answer

  Following your suggestion, we examined the expression of other claudin subtypes. We found that claudin-1, 4, 7 and 12 may be expressed in HaCaT cells (Supplementary figure S9). Other claudins without claudin-4 may be involved in the recovery of TER after 48 h of UVB exposure. We discussed it in the Discussion. Please see line 281.

Comment 6

The authors state that ZO-1 is unchanged in their experiments. However, they only show 0 and 6 h. All the timepoints should be presented as shown for claudin-1 and -4.

Answer

  Following your suggestion, we showed the images of ZO-1 at all the timepoints. Please see new figure 2.

Comment 7

The authors should mention in the discussion that the 5 mJ/cm² dose of UVB does not cause an increase of ROS, as they show in their recent paper. It would make it easier to follow without having to look it up.

Answer

  Following your suggestion, we mentioned in the discussion that the 5 mJ/cm² dose of UVB does not cause an increase of ROS. Please see line 287.

Comment 8

The text needs some rephrasing and clearer formulation, especially in the introduction.

Answer

  Following your suggestion, we check the rephrase and mistakes of grammar.

Round 2

Reviewer 1 Report

The manuscript has been much improved. Are the images in figure S7 after UBV irradiation? The authors claim that claudin-1 localization at tight junctions is changed by UVB, so the authors should evaluate the colocalization of claudin-1 and ZO-1 with z axis image before and after UVB irradiation. Other than that, I would agree to publish with the current edition.

Author Response

Comment 1

Are the images in figure S7 after UBV irradiation? The authors claim that claudin-1 localization at tight junctions is changed by UVB, so the authors should evaluate the colocalization of claudin-1 and ZO-1 with z axis image before and after UVB irradiation. Other than that, I would agree to publish with the current edition.

Answer

  The images in figure S7 were obtained just before UVB irradiation. The colocalization of CLDN1 with ZO-1 was shown in new figure S8. The images were taken near the apical membrane. CLDN1 was colocalized with ZO-1 at 0 h. CLDN1 disappeared after 6 h of UVB irradiation without affecting the localization of ZO-1. We tried to take images of colocalization of CLDN1 with ZO-1 in xz section. However, we could not obtain the data of xz section because the signal of ZO-1 was very weak. There is a possibility that CLDN1 was distributed in not only the cytosol, but also diffusively the lateral membrane. Therefore, we modified the manuscript. Please see line 335.

Reviewer 2 Report

The revised manuscript of Kobayashi and colleagues has taken most of the requested points in account. However, some of them were only improved in parts and new minor points have been arisen, which should be clarified.

Answer 1, Comment:
Thank you for providing the lateral view in Figure S7. However, the authors should not only show one timepoint if they like to show a difference or not. The control should be shown, too. Or is this the control condition? In addition, there is still no counterstaining presented, which would allow also to discriminate whether Cldn1 and ZO-1 are only apically located or within the tight junction. For Cldn1 under several conditions a shift out of the tight junction has been described in literature. Although the location was still in apical regions.

Answer 2, Comment:
The description is still not clear enough. Is 5 mJ/cm² the total dose? How long were the cells treated at time point 0 with which dose?

Answer 3, Comment:

Fig. 1A still only shows cell viability at 0,3,6, and 24 h. The authors maybe accidentally removed the new figure instead of the old?

Answer 4, Comment:

Ok.

Answer 5, Comment:

Supplementary Figure S9: What are the units? Was there an expression correction against an internal control like a housekeeping gene? What would be expression = 1? Furthermore, the description of the method should be added.

Answer 6, Comment:

Thank you for adding the stainings. For completeness, please add the blots and relative protein expression levels, too.

Answer7, Comment:

Ok.

Answer 8, Comment:

Some rephrasing made the text less concrete, for example, it is wrong to explain that the apical side of the lateral membrane is called tight junction (line 89). The tight junction is a cell-cell-contact that is located at apical region of the lateral membranes.

Further minor:

  • The explanation makes no sense: “The TER values were about 328.8±40.6 (Ω∙cm2) in HaCaT cells, which indicate so called leaky membrane. Therefore, the transcellular pathway may not contribute to the reduction of TER.” How can the authors only having the TER make any statements about the transcellular pathway?
  • Supplementary Fig. S8: Full blots are missing.

Author Response

We thank you very much for your careful reading of our manuscript and valuable comments.

Comment 1

Thank you for providing the lateral view in Figure S7. However, the authors should not only show one timepoint if they like to show a difference or not. The control should be shown, too. Or is this the control condition? In addition, there is still no counterstaining presented, which would allow also to discriminate whether Cldn1 and ZO-1 are only apically located or within the tight junction. For Cldn1 under several conditions a shift out of the tight junction has been described in literature. Although the location was still in apical regions.

Answer

  The images in figure S7 were obtained just before UVB irradiation. The colocalization of CLDN1 with ZO-1 was shown in new figure S8. The images were taken near the apical membrane. CLDN1 was colocalized with ZO-1 at 0 h. CLDN1 disappeared after 6 h of UVB irradiation without affecting the localization of ZO-1. We tried to take images of colocalization of CLDN1 with ZO-1 in xz section. However, we could not obtain the data of xz section because the signal of ZO-1 was very weak. There is a possibility that CLDN1 was distributed in not only the cytosol, but also diffusively the lateral membrane. Therefore, we modified the manuscript. Please see line 335.

Comment 2

The description is still not clear enough. Is 5 mJ/cm² the total dose? How long were the cells treated at time point 0 with which dose?

Answer

Following your suggestion, we described the detail of UVB treatment. Please see line 372.

Comment 3

Fig. 1A still only shows cell viability at 0,3,6, and 24 h. The authors maybe accidentally removed the new figure instead of the old?

Answer

  We showed cell viability at 0,3,6, and 48 h. Please see new figure 1A and line 82.

Comment 4

Supplementary Figure S9: What are the units? Was there an expression correction against an internal control like a housekeeping gene? What would be expression = 1? Furthermore, the description of the method should be added.

Answer

  Following your suggestion, we described the method. Please see the legend of new supplementary figure S10.

Comment 5

Thank you for adding the stainings. For completeness, please add the blots and relative protein expression levels, too.

Answer

  At present, we focus on the expression and function claudin-1. TER values were transiently decreased after 6 h of UVB irradiation. We clearly showed the tight junctional localization of claudin-1 was decreased after 6 h of UVB irradiation, whereas that of ZO-1 was not in the new figures 1 and 2. Therefore, we decided that the blots of ZO-1 in all time points are unnecessary.

Comment 6

Some rephrasing made the text less concrete, for example, it is wrong to explain that the apical side of the lateral membrane is called tight junction (line 89). The tight junction is a cell-cell-contact that is located at apical region of the lateral membranes.

Answer

  Following your suggestion, we corrected them.

Comment 7

The explanation makes no sense: “The TER values were about 328.8±40.6 (Ω∙cm2) in HaCaT cells, which indicate so called leaky membrane. Therefore, the transcellular pathway may not contribute to the reduction of TER.” How can the authors only having the TER make any statements about the transcellular pathway?

Answer

  We corrected the sentence. Please see line 280.

Comment 8

Supplementary Fig. S8: Full blots are missing.

Answer

  Following your suggestion, we showed the full blots. Please see new supplementary figure S11.